# Performance of *CADM1, MAL* and *miR124-2* methylation as triage markers for early detection of cervical cancer in self-collected and clinician-collected samples: an exploratory observational study in Papua New Guinea

Monica Molano ,[1,2] Dorothy A Machalek,[1,3] Grace Tan,[4] Suzanne Garland,[5,6] Prisha Balgovind,[2,5] Gholamreza Haqshenas,[5,6] Gloria Munnull,[7] Samuel Phillips,[2,5] Steven G Badman,[3] John Bolnga ,[7,8] Alyssa Marie Cornall,[2,5] Josephine Gabuzzi,[7,8] Zure Kombati,[9] Julia Brotherton,[4,10] Marion Saville,[4] David Hawkes,[4] John Kaldor,[3] Pamela Josephine Toliman,[11] Andrew John Vallely ,[3,12] Gerald L Murray[1,6]

AJV and GLM are joint senior authors.

**Correspondence to**
Dr Monica Molano;
monica.molanoluque@thewomens.org.au

## ABSTRACT

**Objective** WHO recommends human papillomavirus (HPV) testing for cervical screening, with triage of high-risk HPV (hrHPV) positive women. However, there are limitations to effective triage for low-resource, high-burden settings, such as Papua New Guinea. In this exploratory study, we assessed the performance of host methylation as triage tools for predicting high-grade squamous intraepithelial lesions (HSIL) in self-collected and clinician-collected samples.

**Design** Exploratory observational study.

**Setting** Provincial hospital, same-day cervical screen-and-treat trial, Papua New Guinea.

**Participants** 44 hrHPV+women, with paired self/clinician-collected samples (4 squamous cell carcinomas (SCC), 19 HSIL, 4 low-grade squamous intraepithelial lesions, 17 normal).

**Primary and secondary outcome measures** Methylation levels of *CADM1, MAL* and *miR124-2* analysed by methylation-specific PCRs against the clinical endpoint of HSIL or SCC (HSIL+) measured using liquid-based-cytology/p16-Ki67 stain.

**Results** In clinician-collected samples, *MAL* and *miR124-2* methylation levels were significantly higher with increasing grade of disease (p=0.0046 and p<0.0015, respectively). *miR124-2* was the best predictor of HSIL (area under the curve, AUC 0.819) while *MAL* of SCC (AUC 0.856). In self-collected samples, *MAL* best predicted HSIL (AUC 0.595) while *miR124-2* SCC (AUC 0.812). Combined *miR124-2/MAL* methylation yielded sensitivity and specificity for HSIL+ of 90.5% (95% CI 69.6% to 98.8%) and 70% (95% CI 45.7% to 88.1%), respectively, in clinician-collected samples, and 81.8% (95% CI 59.7% to 94.8%) and 47.6% (95% CI 25.7% to 70.2%), respectively, in self-collected samples. *miR124-2/MAL* plus HPV16/ HPV18 improved sensitivity for HSIL+ (95.2%, 95% CI 76.2% to 99.9%) but decreased specificity (55.0%, 95% CI 31.5% to 76.9%).

**Conclusion** *miR124-2/MAL* methylation is a potential triage strategy for the detection of HSIL/SCC in low-income and middle-income country.

## STRENGTHS AND LIMITATIONS OF THIS STUDY

⇒ The assessment of methylation markers in self-collected vaginal samples recollected in a low-middle-income country allows the detection of high-grade squamous intraepithelial lesion (HSIL+) and open the way to full molecular screening in these settings.

⇒ A direct comparison of self-collected samples and paired clinician-collected samples was possible, thus identifying different performance and optimal cut-off values by sample type.

⇒ Study design allowed exploration of the clinical performance of extended genotyping (human papillomavirus 16/18/31/33/45/52/58) combined with host methylation analysis for the detection of HSIL.

⇒ A small sample size led to broad CI.

⇒ Due to limited specialist staff and infrastructure, liquid-based cytology was the diagnostic reference standard rather than a histological endpoint.

## INTRODUCTION

Cervical cancer is the fourth most frequently diagnosed cancer and the fourth leading cause of cancer-related death in women, with an estimated 604 000 new cases and 342 000 deaths worldwide in 2020.[1] Age-standardised rates are disproportionally higher in

low-income and middle-income countries (LMICs) than in high-income countries (18.8 vs 11.3 per 100 000 for incidence; 12.4 vs 5.2 per 100 000 for mortality), primarily due to disparities in access to effective prevention, screening and treatment strategies.[2–5] Infection with high-risk human papillomavirus types (hrHPV) may cause cervical abnormalities graded histologically as cervical intraepithelial neoplasia (CIN, grades 1–3) or cancer (principally squamous cell carcinoma (SCC) or adenocarcinoma), or graded cytologically (using exfoliated cells from the cervix) as low-grade squamous intraepithelial lesion (LSIL), high-grade squamous intraepithelial lesion (HSIL) or SCC.[6]

To achieve cervical cancer elimination,[5] many countries are transitioning their cervical screening programmes to HPV nucleic acid-based testing, which is more sensitive than cytology for detecting underlying high-grade disease. Furthermore, sample collection can be self-administered, which has been shown to increase screening participation in many settings.[6–11] Another important advantage is that nucleic acid amplification testing and subsequent treatment can be performed at point-of-care, greatly simplifying programme delivery, particularly in low-resource settings. These advances led the WHO, in 2021, to endorse HPV screen-and-treat guidelines for women in LMIC.[6]

An important limitation of HPV testing is that it suffers from a lower specificity than cytology, leading to concern that the treatment of all hrHPV+ women would result in substantial overtreatment.[12] Hence, a second (triage) test is recommended to identify hrHPV+ women at highest risk of having underlying disease requiring treatment. The triage technologies most used are cytology, partial genotyping for HPV16 or 18, and in many low-resource settings, visual inspection using acetic acid or Lugol's iodine.[12] However, these triage approaches present challenges for sensitivity and/or specificity in LMIC settings.[8 12]

Molecular triage tests are advantageous over current methods as they are less subjective and could be performed on multiple specimen types, including self-samples. In particular, aberrant host gene DNA methylation patterns in promoter regions of some genes have shown promise as a triage marker in cervical screening in high-income countries. Aberrant methylation is pivotal in cervical carcinogenesis; it causes changes in gene expression, faulty condensation and chromosomal instability.[13 14] Increased DNA methylation has been associated with hrHPV persistence[13] and correlates with increasing severity of cervical disease.[14 15] DNA methylation markers such as *CADM1, MAL* and *miR124-2* have shown good performance for the detection of CIN2 or worse (CIN2+) among hrHPV+women.[14 16 17] However, data on their performance have been based on studies conducted in high-income settings and among highly screened populations.[18] There are currently no data on the performance of these markers among LMIC populations by using self-collected vaginal samples, nor in settings where

HPV-based screening and same-day treatment strategies are used.

Papua New Guinea has a high burden of cervical cancer, with age-standardised incidence and mortality rates of 29.2 and 19.1 per 100 000, respectively, making it the second most common cancer among Papua New Guinean women.[3] A trial study (HPV self-collect, test and same-day-treat, HPV-STAT) performed on women from this country evaluated the clinical performance, treatment completion rates, adverse events profile and acceptability of a fully integrated screening strategy, comprising point-of-care HPV test of self-collected specimens and same-day thermal ablation.[19] This trial supported the introduction of HPV screening and same-day treatment for the control of cervical cancer in LMIC.[19] In the current study, we explored the performance of *CADM1, MAL* and *miR124-2* DNA methylation markers alone and in combination with HPV16/18 and extended genotyping (HPV16/18/31/33/45/52/58) as a triage strategy for the detection of HSIL+ among HPV+women that participated in this cervical screening trial. A direct comparison was also performed for each methylation marker, between paired self-collected and clinician-collected samples.

## MATERIALS AND METHODS

### Study population and design of the trial

The exploratory study was conducted as a substudy of HPV-STAT a prospective, single-arm intervention trial among women attending cervical screening services at two clinical sites in Papua New Guinea. The trial is registered with ISRCTN, ISRCTN13476702 (https://www.isrctn.com/editorial/retrieveFile/3ed9173e-cce5-4158-a586-577775f0cbdd/35731). Study design, recruitment and protocols have been described.[19] Briefly, between 5 June 2018 and 6 January 2020, 4285 women aged 30–59 years attending clinics for routine cervical screening at Modilon Hospital (Madang Province) and Mt Hagen General Hospital (Western Highlands Province), who meet the inclusion criteria were invited to participate. Women who were pregnant or who had given birth in the past 6 weeks, or women who had a history of cervical cancer or hysterectomy, were excluded.[19] Those women able to provide written informed consent were eligible to participate, enrolled sequentially and collected a vaginal specimen using a cytobrush ('Just for Me', Preventative Oncology International, Cleveland Heights, Ohio, USA), which was placed into a 20 mL ThinPrep vial (Hologic, Marlborough, Massachusetts, USA).[19]

### Point-of-care HPV testing and follow-up assessment

For self-collected vaginal specimens, a 1 mL aliquot was tested for the presence of oncogenic HPV genotypes using the Xpert HPV Test (GeneXpert; Cepheid, Sunnyvale, California, USA) as per the manufacturer's instructions. Results were reported as 'HPV16', 'HPV18/45' and 'other HPV' (a summary result for HPV31, 33, 35, 39, 51, 52, 56, 58, 59, 66 and 68) and were provided to women

before midday to allow sufficient time for same-day counselling, pelvic examination and treatment/referral as indicated.[19] All women with a negative HPV test were informed of their results and advised to return to the clinic for HPV-based screening in 5 years.

hrHPV+women provided a cervical specimen collected by a clinician using a Cervex-Brush Combi (Rovers Medical Devices, Oss, The Netherlands), placed in a 20 mL ThinPrep vial and stored at 4°C prior to shipment to the Australian Centre for the Prevention of Cervical Cancer in Melbourne, Australia for liquid-based cytology (LBC) and p16/Ki67 dual stain cytology. A 15% random sample of HPV-negative women were also asked to provide a clinician-collected cervical specimen for LBC, as above.

### Cytological assessment

LBC was carried out in accordance with established laboratory procedures at the Australian Centre for the Prevention of Cervical Cancer. Slides were independently assessed by two cytologists blinded to HPV test results. Where both cytologists agreed on a diagnosis of HSIL or HSIL+, a final diagnosis was recorded. If the assessment differed, dual p16/Ki-67 dual stain was carried out by using CINTec PLUS Cytology (Roche Diagnostics, Pleasanton, USA) to make a final diagnosis.[19] The primary clinical endpoint was then HSIL or SCC (HSIL+) using LBC/p16-Ki67 stain. LBC was accepted as a reference standard rather than histology, which is the gold reference in high-resource settings. It was not feasible from a staffing and logistical perspective to provide colposcopy or to collect cervical biopsies for histological examination in Papua New Guinea.

### Participants in the exploratory study and design

The substudy included 44 paired cervical and vaginal samples from women participating in the trial at Mt Hagen General Hospital (Mount Hagen, Western Highlands Province). This included all 23 hrHPV+HSIL+ (19 HSIL and 4 SCC) cases identified on LBC by the end of 2018 and 21 randomly selected hrHPV+normal/LSIL samples (17 normal LBC and 4 LSIL). Sample size selection was driven by a published study by Li et al that showed through simulation studies and real data from two studies downloaded from the NCBI Gene Expression Omnibus that at least 12 specimens in each group is needed to detect truly differential DNA methylation with enough power (≥ 80%), reproducible data and consistency when using different statistical methods.[20] Molecular biologists and technicians were blinded to point-of-care HPV and clinical diagnosis.

### DNA extraction and HPV-specific typing

DNA was extracted as described previously, quantitated by Qubit Fluorometer (Life Technologies, California, USA) and assessed for integrity by quantitative PCR amplification of a 260 base-pair product of the human beta-globin gene.[21] In addition, DNA extracted from cervical cell line SiHa (1–2 copies of HPV16 per cell,

American Type Culture Collection (ATCC) Cat# HTB-35, RRID:CVCL_0032; ATCC, Manassas, Virginia, USA; 100 ng) was used as positive control for methylation analysis. This DNA was confirmed to have only HPV16 by HPV typing for 28 HPV types (Anyplex II HPV28 Detection, Seegene, Seoul, South Korea).

HPV genotyping was performed using the Anyplex II HR HPV Detection multiplex assay (Seegene, Seoul, South Korea), which detects 14 oncogenic HPV genotypes (16, 18, 31, 33, 35, 39, 45, 51, 52, 56, 58, 59, 66, 68) and an internal control, according to manufacturer's recommendations.

### Bisulphite DNA modification and quantitative methylation-specific PCR for *CADM1*, *MAL* and *miR124* genes

DNA from the paired samples (1–100 ng) and control (SiHa, 100 ng) were bisulphite treated using Methylamp DNA modification Kit (Epigentek, Brooklyn, New York, USA) as per the manufacturer's instructions. Modified DNA was eluted in 40 µL of the methylamp elution buffer. Samples without the adequate concentration were excluded from the bisulphite modification and methylation analysis.

Quantitative methylation-specific PCR (qMSP) targeting CpG sites in promoter regions of *CADM1* (promoter region M9), *MAL* (promoter region M1) and *miR124-2* (promoter region 2) was performed as previously described,[14] with minor modifications. Due to the use of a different platform (Light Cycler 480 II, Roche), validations of the qPCRs were performed utilising duplicated dilution series of SiHa cell line (intra and inter assays of reproducibility) and by using a training panel of samples as reported previously.[21] As a reference, a qMSP for the housekeeping gene b-actin (*ACTB*) was performed. Samples with Ct values for *ACTB* of >34 were considered *ACTB* negative as this indicated poor sample quality due to insufficient DNA or inadequate bisulphite conversion. Target DNA methylation levels were normalised to the reference gene as described previously.[21]

### Statistical analysis

The percentage of methylation was calculated as described previously.[22] Differences in the median percentage methylation by disease grade (SCC, HSIL, LSIL/normal) were visualised using box and whisker plots and compared using Wilcoxon test (between groups) and Kruskal-Wallis (overall analysis).[20 21] Area under the receiver operating characteristic curve (ROC) (AUC) was used to assess the ability of the methylated genes to classify HSIL or SCC, providing two main outcomes: the diagnostic accuracy of the test and the optimal cut-off point value for dichotomisation of the methylation test.[23] The optimal cut-off point was calculated by using the maximum sum of sensitivity and specificity.[24] Sensitivity (number of correct positives (ie, positive for at least one marker)/number of reference assay positives) and specificity (number of correct negatives/number of reference assay negatives) were determined for the following triage strategies: (1) single

gene methylation analysis, (2) combination of methylation markers, (3) HPV 16 genotyping, (4) combination of HPV16/18 genotyping, (5) combination of HPV 16/18/31/33/45/52/58 genotyping, (6) HPV16/18 with single and combined methylation analysis and (7) HPV 16/18/31/33/45/52/58 with single and combined methylation analysis.

Median percentage methylation for each marker was compared for self-collected and clinician-collected vaginal paired specimens using the Wilcoxon signed rank test. The vaginal specimen was considered the test while the cervical specimen acted as the non-reference standard. Agreements between DNA methylation results from cervical and vaginal specimens were determined using the optimal cut-off point established for each gene by calculating positive percent agreement (PPA), negative percent agreement (NPA) and overall per cent agreement (OPA) as per recommendations by the Food and Drug Administration, USA.[25]

The results were analysed by using XLSTAT and the statistical platform R studio (V.4.0.1) and programmes ggplots2 (V.3.3.2), ggpubr (V.0.4), pROC (V.1.16.2) and cutpointr (V.1.1.1).[21 24]

We used the Standards for Reporting of Diagnostic Accuracy Studies (STARD) reporting guidelines for reporting diagnostic accuracy studies.[26]

### Patient and public involvement
There was no patient or public involvement in study design or conduct.

## RESULTS
### DNA methylation of *CADM1, MAL* and *miR124-2* and associations with lesion grade
Overall, 44 hrHPV+paired cervical and vaginal samples from women participating in the trial at Mt Hagen General Hospital were used (23 cases (19 HSIL and 4 SCC 8) and 21 normal/LSIL (17 normal LBC and 4 LSIL)). Of these samples, 41 (93.2%) clinician-collected (4 SCC, 17 HSIL, 4 LSIL and 16 normal LBC) and 43 (97.7%) self-collected samples (4 SCC, 18 HSIL, 4 LSIL and 17 normal LBC) were deemed assessable. The methylation analysis was not used for the clinical management of the women and thus did not induce any adverse effects.

In clinician-collected samples DNA methylation of *MAL* was higher with increasing disease grade from normal/LSIL samples (1.4%; 95% CI 0.8% to 3.1%), HSIL (median 2.0%; 95% CI 0.7% to 20.4%) and SCC (63.3%; 95% CI 46.8% to 67.9%) (p=0.0046) (figure 1, top central panel). Methylation of *mirR124-2* was also higher with increasing disease grade from normal/LSIL samples (2.6%; 95% CI 1.1% to 5.1%), HSIL (5.6%; 95% CI 3.9% to 20.5%) and SCC (24.1%; 95% CI 12.4% to 34.5%) (p=0.0015) (figure 1, top right panel). Methylation of *CADM1* did not differ significantly with increasing cytology grade (figure 1, top left panel).

Using clinician-collected samples, ROC analysis showed that *miR124-2* was the best methylation marker to distinguish SCC and HSIL (from normal/LSIL), with an AUC of 0.850 and 0.819, respectively (figure 2A,C). Methylation of *MAL* showed higher AUC for detecting SCC (0.856) than HSIL (0.700). *CADM1* methylation showed poor performance for detecting SCC or HSIL with AUC of 0.700 and 0.607, respectively (figure 2A,C).

In self-collected samples, there was a trend of increasing DNA methylation with disease grade for *MAL* and *miR124-2*, though this did not reach significance (p=0.083, p=0.075, respectively) (figure 1, lower centre panel). As found for clinician-collected samples, methylation of *CADM1* did not differ significantly with increasing cytology grade (figure 1, lower left panel).

ROC analysis showed that from self-collected samples, *miR124-2* was the best methylation marker to distinguish SCC, but not HSIL, from normal/LSIL with an AUC of 0.812 and 0.563, respectively (figure 2B,D). Methylation of *MAL* or *CADM1* showed a lower clinical performance for SCC (AUC of 0.725 and 0.750, respectively) than *miR124-2*, and a similar performance for HSIL (AUC of 0.595 and 0.515, respectively) (figure 2B,D).

### Diagnostic performance of DNA methylation markers for the detection of HSIL or SCC (HSIL+)
Diagnostic performance of individual DNA methylation markers for detecting HSIL+ in clinician-collected samples showed that *miR124-2* methylation had the highest sensitivity of 81.0% (95% CI 58.1% to 94.6%) and *MAL* methylation the highest specificity of 90.0% (95% CI 68.3% to 98.8%) (table 1). A combination of *miR124-2* and *MAL* showed the optimal diagnostic performance with a sensitivity of 90.5% (95% CI 69.6% to 98.8%) and specificity of 70.0% (95% CI 45.7% to 88.1%). A combination of *miR124-2/MAL* and *CADM1* showed a maximal sensitivity of 100% (95% CI 83.9% to 100%) at the expense of a low specificity, 25% (95% CI 8.7% to 49.1%).

In self-collected samples, *miR124-2* methylation had a sensitivity of 77.3% (95% CI 54.6% to 92.2%) for detecting HSIL+, but a much lower specificity 47.6% (95% CI 25.7% to 70.2%). *MAL* showed the best specificity 90.5% (95% CI 69.6% to 98.8%) at the expense of a low sensitivity 32.0% (95% CI 13.9% to 54.9%) (table 1). A combination of *miR124-2* and *MAL* improved the diagnostic performance with a sensitivity of 81.8% (95% CI 59.7% to 94.8%) and specificity of 47.6% (95% CI 25.7% to 70.2%). A combination of *miR124-2/MAL* and *CADM1* increased the sensitivity to 90.9% (95% CI 70.8% to 98.9%) at the expense of a low specificity of 23.8% (95% CI 8.2% to 47.2%) (table 1).

### Diagnostic performance of HPV genotyping combined with DNA methylation markers for the detection of HSIL or SCC (HSIL+)
Diagnostic performance of HPV16 for detecting HSIL+ in clinician-collected samples showed a sensitivity of 52.2% (95% CI 30.6% to 73.2%) and specificity of 81.0% (95% CI

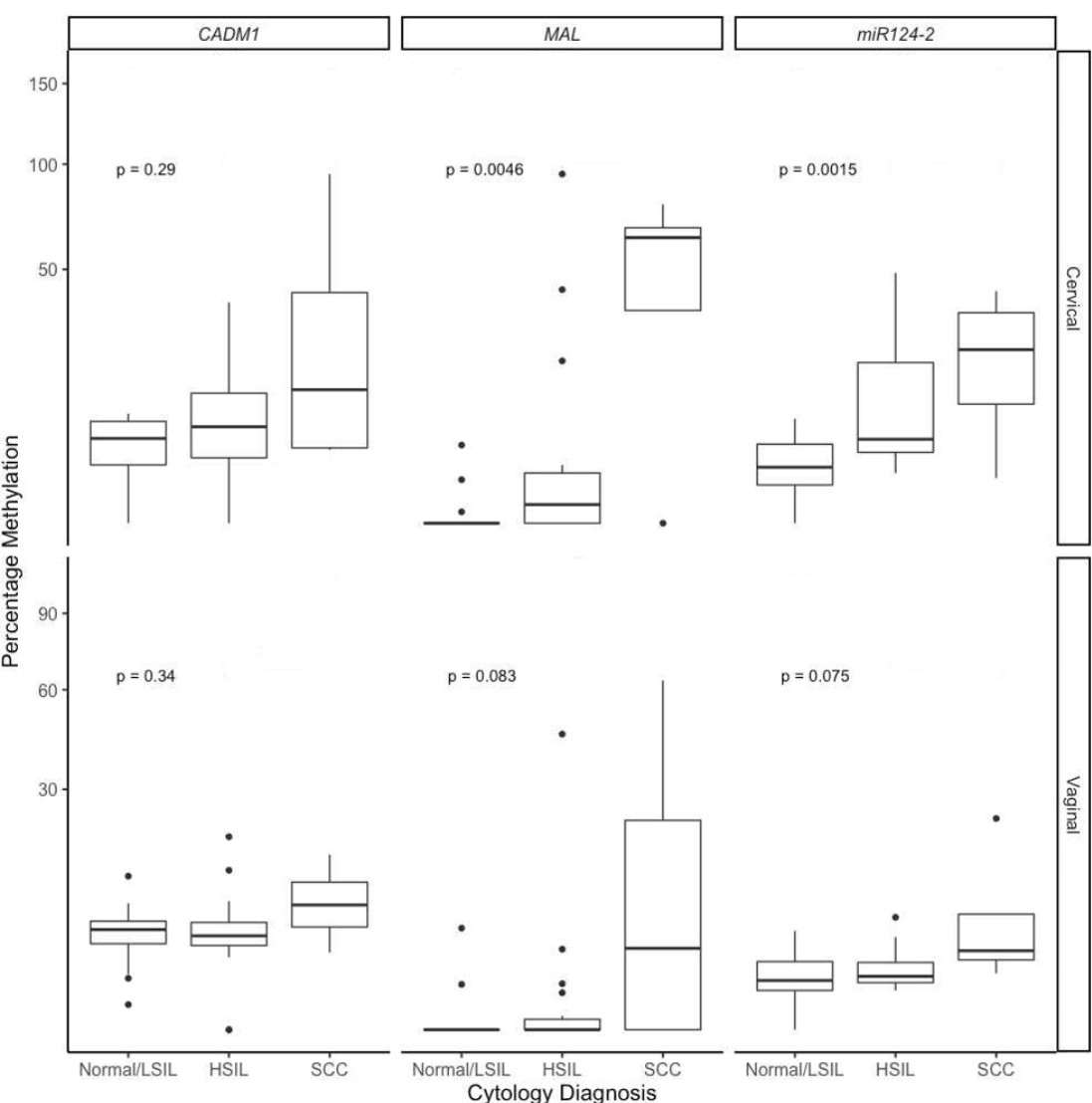

**Figure 1** Percentage of DNA methylation of gene *CADM1, MAL* and *miR124-2* according to cytological grade in oncogenic HPV positive clinician-collected cervical samples (cervical) in the top panel and self-collected vaginal samples (vaginal) in the bottom panel. Whiskers correspond to the 1st and 3rd quartiles (the 25th and 75th percentiles). Overall significance by Kruskal-Wallis is indicated. HPV, human papillomavirus; HSIL, high-grade squamous intraepithelial lesions; LSIL, low-grade squamous intraepithelial lesions; SCC, squamous cell carcinomas.

58.1% to 94.6%) (table 2). HPV16 and HPV18 combined showed a sensitivity of 60.9% (95% CI 38.5% to 80.3%) and specificity of 81.0% (95% CI 58.1% to 94.6%). A combination of *miR124-2/MAL* methylation markers and HPV16/18 genotyping showed a satisfactory diagnostic performance with a sensitivity 95.2% (95% CI 76.2% to 99.9%) and specificity of 55% (95% CI 31.5% to 76.9%). A combination of *MAL/CADM1* and HPV16/18 genotyping showed a maximal sensitivity of 100% (95% CI 85.9% to 100%) at the expense of a specificity of 30% (95% CI 11.9% to 54.3%). Using HPV16/18/31/33/45/52/58 genotyping, either as a single marker or combined with different methylation markers, showed high sensitivities but at the expense of low specificities (table 2).

For self-collected samples, the detection of individual and combined HPV genotypes had the same diagnostic performance as seen in clinician-collected samples (table 2). A

combination of *miR124-2* or *miR124/MAL* and HPV16/HPV18 showed similar sensitivity found for clinician-collected samples of 95.5% (95% CI 77.2% to 99.9%), but reduced specificity (33.3%, 95% CI 14.6% to 50.7%). A combination of *miR124-2/CADM1* and HPV16/18 genotyping showed a maximal sensitivity of 100% (95% CI 84.6% to 100%) and specificity of 23.8% (95% CI 8.2% to 47.2%). Adding HPV16/18/31/33/45/52/58 genotyping combined with different methylation markers showed high sensitivities at the expense of low specificities (table 2).

### Comparison of levels of methylation between paired self-collected and clinician-collected samples

There was generally a lower percentage of methylation for each marker in vaginal samples compared with their paired cervical samples (figure 3). However, this only

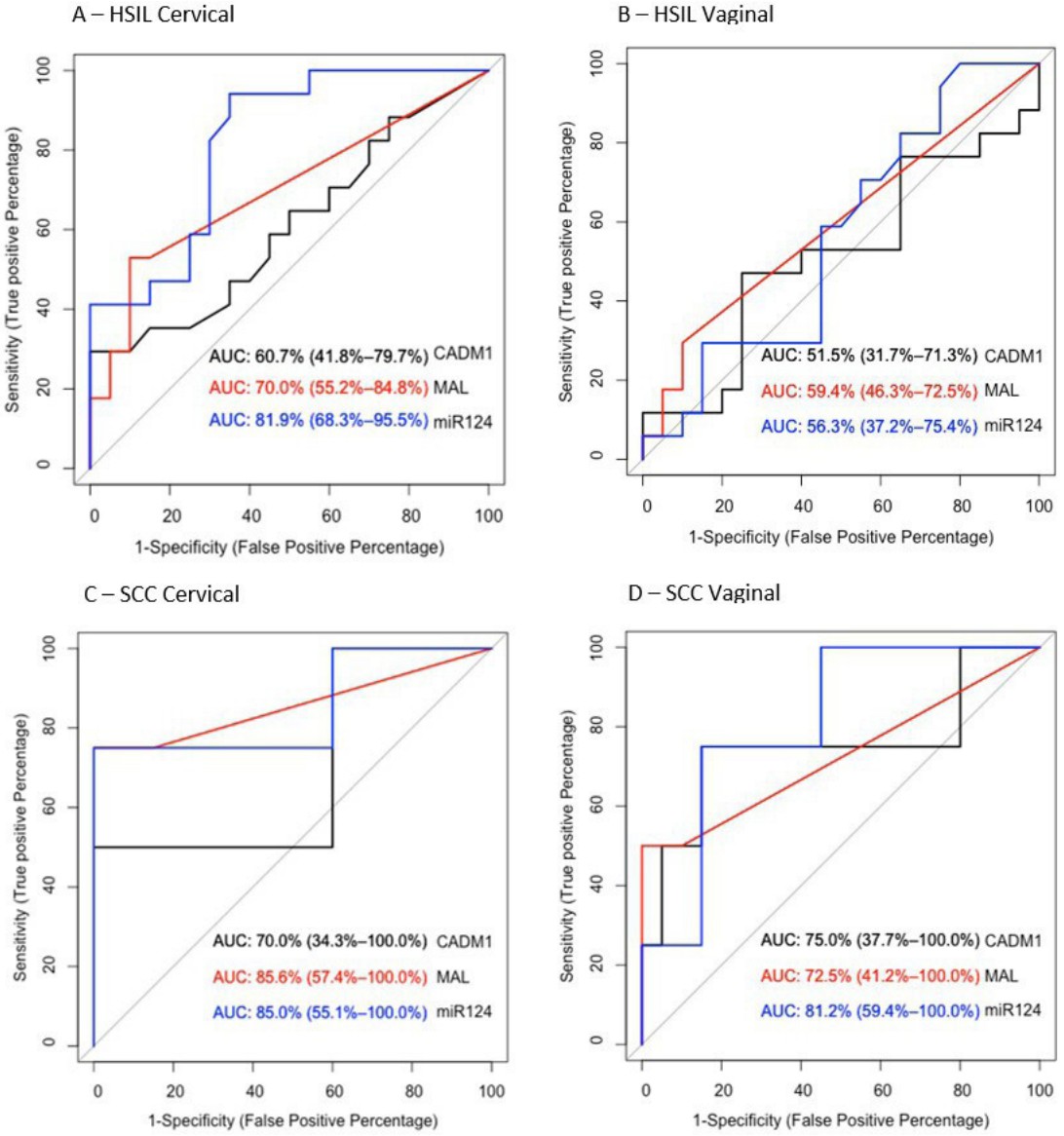

**Figure 2** Receiver operating characteristic (ROC) curve values of the performance of DNA methylation of *CADM1, MAL* and *miR124-2* for distinguishing HSIL from LSIL/normal, and SCC from LSIL/normal, stratified by type; clinician-collected cervical samples (A, C) or self-collected vaginal samples (B, D). HSIL, high-grade squamous intraepithelial lesions; LSIL, low-grade squamous intraepithelial lesions; SCC, squamous cell carcinomas.

reached statistical significance for *miR124-2* in HSIL (p=0.0002) (figure 3).

### Agreement of paired self-collected and clinician-collected samples for *miR124-2, MAL* and *CADM1* methylation

Analysis was performed for 40 patients with paired, assessable samples. Initial analysis was performed without considering the underlying disease status. OPA, PPA and NPA were 62.5%, 77.3% and 44.4%, respectively, for *miR124-2* methylation, 77.5%, 46.2% and 92.6%, respectively, for *MAL* methylation, and 57.5%, 67.9% and 33.3%, respectively, for *CADM1*.

In analyses limited to HSIL+paired clinician-collected and self-collected samples, the OPA, PPA and NPA were 75.0% (95% CI 53.1% to 88.8%), 81.3% (95% CI 57.0% to 93.4%) and 50.0% (95% CI 15.0% to 85.0%),

respectively, for *miR124-2* methylation, 65.0% (95% CI 43.3% to 81.9%), 45.5% (95% CI 21.3% to 72.0%) and 88.9% (95% CI 56.5% to 98.0%), respectively, for *MAL* methylation; and 65.0% (95% CI 43.3% to 81.9%), 68.8% (95% CI 44.4% to 85.8%) and 50.0% (95% CI 15.0% to 85.0%), respectively, for *CADM1*.

### DISCUSSION

In this study, we explored *CADM1, MAL* and *miR124-2* DNA methylation using clinician-collected and self-collected HPV+samples from Papua New Guinea, a setting with a high burden of cervical disease. In clinician-collected cervical samples, DNA methylation of *MAL* and *miR124-2* was significantly higher in HSIL/cancer compared with normal/LSIL samples while the methylation of *CADM1*

**Table 1** Performance of host DNA methylation markers for detection of HSIL+

| Sample type | Gene* | Per cent methylation cut-off | Sensitivity % (95% CI) | Specificity % (95% CI) |
|---|---|---|---|---|
| Cervical, clinician collected | miR124-2 | 3.7 | 81.0 (58.1 to 94.6) | 70.0 (45.7 to 88.1) |
| | MAL | 0.25 | 57.1 (34.0 to 78.2) | 90.0 (68.3 to 98.8) |
| | CADM1 | 3.5 | 80.9 (58.1 to 94.6) | 40.0 (19.1 to 63.9) |
| | miR124-2/MAL | 3.7/0.25 | 90.5 (69.6 to 98.8) | 70.0 (45.7 to 88.1) |
| | miR124-2/CADM1 | 3.7/3.5 | 95.2 (76.2 to 99.9) | 25.0 (8.7 to 49.1) |
| | MAL/CADM1 | 0.25/3.5 | 95.2 (76.2 to 99.9) | 35.0 (15.4 to 59.2) |
| | miR124-2/MAL/CADM1 | 3.7/0.25/3.5 | 100 (83.9 to 100) | 25.0 (8.7 to 49.1) |
| Vaginal, self-collected | miR124-2 | 1.2 | 77.3 (54.6 to 92.2) | 47.6 (25.7 to 70.2) |
| | MAL | 0.1 | 32.0 (13.9 to 54.9) | 90.5 (69.6 to 98.8) |
| | CADM1 | 3.9 | 68.2 (45.1 to 86.1) | 33.3 (14.6 to 57.0) |
| | miR124-2/MAL | 1.2/0.1 | 81.8 (59.7 to 94.8) | 47.6 (25.7 to 70.2) |
| | miR124-2/CADM1 | 1.2/3.9 | 86.4 (65.1 to 97.1) | 23.8 (8.2 to 47.2) |
| | MAL/CADM1 | 0.1/3.9 | 81.8 (59.7 to 94.8) | 28.6 (11.3 to 52.2) |
| | miR124-2/MAL/CADM1 | 1.2/0.1/3.9 | 90.9 (70.8 to 98.9) | 23.8 (8.2 to 47.2) |

Shaded rows show markers or a combination of markers with the best diagnostic performance and those with the maximal sensitivity for detection of HSIL+.
*Positive for at least one of the indicated host genes.
HSIL, high-grade squamous intraepithelial lesion.

failed to distinguish between cytology grades. Methylation of *miR124-2* showed the highest AUC to distinguish HSIL from normal/LSIL and the second-highest performance to predict SCC, with AUC values above 0.819. Exploration of diagnostic performance of *miR124-2* in combination with *MAL* for the detection of HSIL+, showed an excellent performance (sensitivity 90.5%, specificity 70%). Similar findings were obtained with self-collected samples, though their performance was reduced. Overall, *miR124-2* in combination with *MAL* DNA methylation performed well in the detection of HSIL+ in women from LMIC. Addition of HPV16/18 or extensive genotyping improved sensitivity at the expense of lower specificity and higher referrals to immediate treatment. *miR124-2/MAL* methylation markers are suitable for further evaluation against histology endpoints, and subsequently in LMIC-based field studies for triage of hrHPV+women to identify high-grade disease.

Increased levels of *miR124-2*, *CADM1* and *MAL* methylation according to lesion grade have been observed in studies performed in clinician-collected cervical samples from high-income populations.[27–29] *miR124-2* methylation alone or in combination with *CADM1* and *MAL* has high sensitivity and specificity for the detection of CIN3+ in HPV+ clinician-collected cervical specimens in women from high-income countries (sensitivity of 68.0%–94.7%, specificity of 50.7%–78.9%).[30]

Few studies have been performed in LMIC using clinician-collected samples. An analysis of the same markers in HPV+ and HIV+ women from Kenya found the AUC of *miR124-2* methylation for CIN2+ and CIN3+

were 0.730 and 0.810, respectively,[31] marginally lower to those obtained in the current exploratory study. However, these results are not directly comparable to ours as their lesion grade was based on histological diagnosis and HIV+ participants. In the current study, we were unable to stratify by HIV status, but assume a low HIV prevalence as the HIV prevalence in the Papua New Guinean general population is <1%. Two other studies performed in South Africa and Thailand (upper-middle-income countries) also showed good performance of some of these markers, alone or in combination, for the detection of HSIL+.[32 33]

*CADM1* was not a good diagnostic marker, this is consistent with the findings of two other studies conducted in high-income countries.[34 35]

No research has been conducted looking methylation in self-collected samples from LMIC. Diagnostic performance of *miR124-2* for detection of HSIL+ in self-collected samples showed a sensitivity of 77.3% and specificity of 47.6% while combining *miR124-2* with *MAL* increased the sensitivity to 81.8%, showing promise for self-sampling applications. While the sensitivity/specificity was slightly lower than clinician-collected samples, this may be offset by higher screening rates achieved by this more acceptable method of collection.[11]

By comparison, studies of *miR124-2* and *MAL* single or in combination with other methylation markers for detection of CIN3+ in self-collected samples from high-income settings, have shown a wide range of sensitivities and specificities with values between 37.5%–78.4% and 47.2%–98%, respectively.[36–39] This wide variation may be explained by variable methodology (self-sampling

**Table 2** Performance of HPV genotyping combined with methylation markers for the detection of HSIL+

| Sample type | HPV genotype * | Host gene† | Percent methylation cut-off | Sensitivity % (95% CI) | Specificity (95% CI) |
|---|---|---|---|---|---|
| Cervical, clinician collected | 16 | NA | NA | 52.2 (30.6 to 73.2) | 81.0 (58.1 to 94.6) |
| | 16,18 | NA | NA | 60.9 (38.5 to 80.3) | 81.0 (58.1 to 94.6) |
| | 16, 18, 31, 33, 45, 52, 58 | NA | NA | 95.7 (78.1 to 99.9) | 33.3 (14.6 to 56.9) |
| | 16,18 | miR124-2 | 3.7 | 85.7 (63.7 to 97.0) | 55.0 (31.5 to 76.9) |
| | 16, 18, 31, 33, 45, 52, 58 | miR124-2 | 3.7 | 95.2 (76-2-99.9) | 25.0 (8.7 to 49.1) |
| | 16,18 | MAL | 0.25 | 76.2 (52.8 to 91.8) | 70.0 (45.7 to 88.1) |
| | 16, 18, 31, 33, 45, 52, 58 | MAL | 0.25 | 100 (84.6 to 100) | 14.3 (3.1 to 36.3) |
| | 16,18 | CADM1 | 3.5 | 90.5 (69.6 to 98.8) | 35.0 (15.4 to 59.2) |
| | 16, 18, 31, 33, 45, 52, 58 | CADM1 | 3.5 | 95.2 (76.2 to 99.9) | 20.0 (5.7 to 43.7) |
| | 16,18 | miR124-2/MAL | 3.7/0.25 | 95.2 (76.2 to 99.9) | 55.0 (31.5 to 76.9) |
| | 16, 18, 31, 33, 45, 52, 58 | miR124-2/MAL | 3.7/0.25 | 100 (83.9 to 100) | 25.0 (8.7 to 49.0) |
| | 16,18 | miR124-2/CADM1 | 3.7/3.5 | 95.2 (76.2 to 99.9) | 25.0 (8.7 to 49.1) |
| | 16, 18, 31, 33, 45, 52, 58 | miR124-2/CADM1 | 3.7/3.5 | 95.2 (76.2 to 99.9) | 15.0 (3.2 to 37.9) |
| | 16,18 | MAL/CADM1 | 0.25/3.7 | 100 (85.9 to 100) | 30.0 (11.9 to 54.3) |
| | 16, 18, 31, 33, 45, 52, 58 | MAL/CADM1 | 0.25/3.7 | 100 (83.9 to 100) | 20.0 (5.7 to 43.7) |
| Vaginal, self-collected | 16 | NA | NA | 52.2 (30.6 to 73.2) | 81.0 (58.1 to 94.6) |
| | 16,18 | NA | NA | 60.9 (38.5 to 80.3) | 81.0 (58.1 to 94.6) |
| | 16, 18, 31, 33, 45, 52, 58 | NA | NA | 95.7 (78.1 to 99.9) | 33.3 (14.6 to 56.9) |
| | 16,18 | miR124-2 | 1.2 | 95.5 (77.2 to 99.9) | 33.3 (14.6 to 50.7) |
| | 16, 18, 31, 33, 45, 52, 58 | miR124-2 | 1.2 | 100 (84.6 to 100) | 14.3 (3.1 to 36.3) |
| | 16,18 | MAL | 0.1 | 68.2 (45.1 to 86.1) | 71.4 (47.8 to 88.7) |
| | 16, 18, 31, 33, 45, 52, 58 | MAL | 0.1 | 95.5 (77.2 to 99.9) | 28.6 (11.3 to 52.2) |
| | 16,18 | CADM1 | 3.9 | 90.9 (70.8 to 98.9) | 33.3 (14.6 to 56.9) |
| | 16, 18, 31, 33, 45, 52, 58 | CADM1 | 3.9 | 100 (84.6 to 100) | 9.5 (1.2 to 30.4) |
| | 16,18 | miR124-2/MAL | 1.2/0.1 | 95.5 (77.2 to 99.9) | 33.3 (14.6 to 57.0) |
| | 16, 18, 31, 33, 45, 52, 58 | miR124-2/MAL | 1.2/0.1 | 100 (84.6 to 100) | 14.3 (3.1 to 36.3) |
| | 16,18 | miR124-2/CADM1 | 1.2/3.9 | 100 (84.6 to 100) | 23.8 (8.2 to 47.2) |
| | 16, 18, 31, 33, 45, 52, 58 | miR124-2/CADM1 | 1.2/3.9 | 100 (84.6 to 100) | 4.8 (0.1 to 23.8) |
| | 16,18 | MAL/CADM1 | 0.1/3.9 | 90.9 (70.8 to 98.9) | 28.6 (11.3 to 52.2) |
| | 16, 18, 31, 33, 45, 52, 58 | MAL/CADM1 | 0.1/3.9 | 100 (84.6 to 100) | 9.5 (1.2 to 30.4) |

Shaded rows show markers or a combination of markers with the best diagnostic performance and those with the maximal sensitivity for detection of HSIL+.
*Detection of at least one of the indicated HPV genotypes.
†Positive for at least one of the indicated host genes.
HPV, human papillomavirus; HSIL, high-grade squamous intraepithelial lesion.

collection devices, methylation detection assays, genes analysed, thresholds to define methylation positivity, study populations, disease endpoints (cytological HSIL or histological CIN2/3+), etc).

Triage combining HPV16/18 genotyping and methylation analysis has shown promise in clinician-collected samples in high-income settings.[37 38 40 41] Methylation analysis of *miR124-2/MAL* combined with HPV16/18 genotyping had a sensitivity for CIN3+ of 77.6% and specificity of 54.8%.[40] A recent prospective cohort study in an upper-middle-income country analysing methylation markers ZNF671/ASTN1/ITGA4/RXFP3/SOX17/DLX1 showed a sensitivity and specificity of 96.6% and 58.3%, respectively, to detect CIN2+ when combining

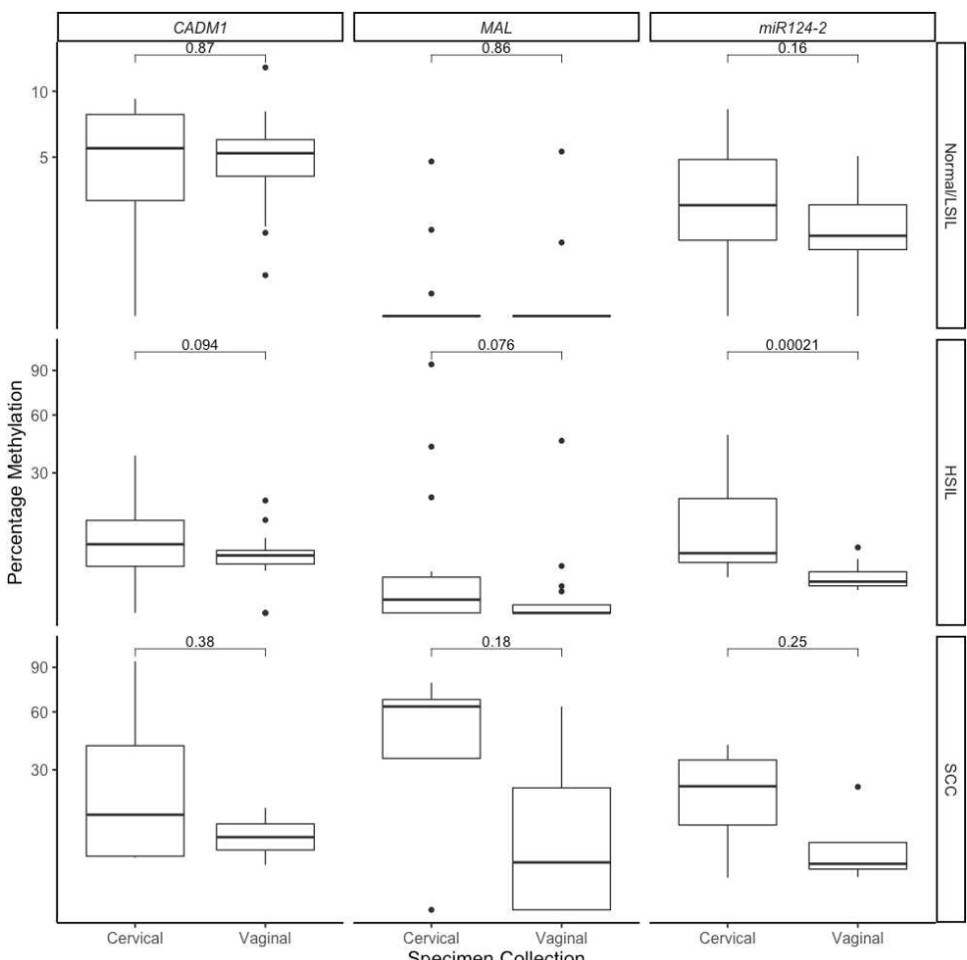

**Figure 3** Percentage methylation of gene *CADM1, MAL* and *miR124-2* comparing paired clinician-collected cervical samples (Cervical) and self-collected vaginal samples (Vaginal). Comparisons assessed by Wilcoxon signed rank test, with whiskers corresponding to the 1st and 3rd quartiles (the 25th and 75th percentiles). HSIL, high-grade squamous intraepithelial lesions; LSIL, low-grade squamous intraepithelial lesions; SCC, squamous cell carcinomas.

methylation and HPV16/18.[42] In our study, combining HPV16/18 detection with *miR124-2/MAL* methylation improved sensitivity (increasing from 90.5% to 95.2%), but reduced specificity (decreasing from 70.0% to 55.0%). In self-collected samples, the sensitivity was similar to clinician-collected samples (95.5%) but the specificity was lower (33.3%).

Combining extended genotyping (HPV16/18/31/33/45/52/58) with miR124-2/*MAL* methylation had similar sensitivity to *miR124/MAL/ HPV16/18* testing but showed lower specificity. Regardless of study population, HPV prevalence, sample type, marker panel and age, the combination of host cell DNA methylation analysis with partial or extended genotyping could serve as a potential triage strategy for hrHPV+ women, at the expense of a higher referral to treatment.

Self-collected samples generally had lower levels of methylation than paired clinician-collected samples, likely due to lower diseased/normal cell ratios. For example, values for the GynTect assay (detecting six methylation markers) were significantly lower for self-collected than clinician-collected samples and the overall concordance

was moderate (kappa 0.394).[43] Further discovery work to identify markers suitable for self-collection should consider using self-collected samples from LMIC. Verlaat *et al* performed a chip-based genome DNA methylation screen on self-samples from a high-income country and identified a three-gene methylation classifier (ASCL1, LHX8 and ST6GALNAC5) that was able to identify CIN3 in cervical self-samples with an ROC of 0.88, which is superior to currently available methods.[44]

Limitations of this study include the small sample size which led to broad CIs; however, it aimed to provide estimated effects (point estimates with CIs) for all measures of interest which are valuable for planning subsequent studies in LMIC. It was not feasible to conduct cervical biopsies in this setting due to limited specialist staff and infrastructure; hence LBC HSIL+ was the diagnostic reference standard rather than a histological endpoint and this may have impacted estimates of sensitivity/specificity for methylation markers. Since HPV testing is more sensitive than cytology at detecting HSIL+, then the selection of cytology as a reference standard will result in an overestimation of the sensitivity of the HPV test and a decrease

in the specificity, compared with results using histology as a reference standard. However, to avoid disease misclassification by cytology all slides were assessed by two independent experienced cytologists, and dual p16/Ki-67 immunostaining was performed to resolve disagreements. A triage test based on methylation markers could be highly suited to regions without access to colposcopy, where the combination of HPV detection, methylation analysis and cytology could confirm HSIL+ in the absence of colposcopy examination. Additional information is required to understand the long-term safety of methylation markers in screening programmes, including longitudinal data from a low-resource setting and analysis of methylation in self-collected samples from the general population and HIV+ women.

## CONCLUSIONS

DNA methylation of *miR124-2/MAL* alone or in combination with HPV16/18 is very promising as a triage test for the prediction of HSIL/SCC among women with oncogenic HPV infections. Self-collected samples showed reduced specificity compared with clinician-collected samples, indicating a need to identify additional biomarkers for this sample type. Future evaluation of these results against histological endpoints, and subsequently in different populations may lead to better tests to manage HPV+ women in LMIC.

**Author affiliations**
¹Centre for Women's Infectious Diseases, The Royal Women's Hospital, Parkville, Victoria, Australia
²Murdoch Children's Research Institute, Parkville, Victoria, Australia
³Kirby Institute, University of New South Wales, Sydney, New South Wales, Australia
⁴Australian Centre for the Prevention of Cervical Cancer, Melbourne, Victoria, Australia
⁵The Royal Women's Hospital, Parkville, Victoria, Australia
⁶Department of Obstetrics, Gynaecology and Newborn Health, The University of Melbourne, Melbourne, Victoria, Australia
⁷Papua New Guinea Institute of Medical Research, Goroka, Papua New Guinea
⁸Obstetrics and Gynaecology, Modilon General Hospital, Madang, Papua New Guinea
⁹Tininga Clinic, Mount Hagen General Hospital, Mount Hagen, Western Highlands Province, Papua New Guinea
¹⁰School of Population Global Health, The University of Melbourne, Melbourne, Victoria, Australia
¹¹Kirby Institute -Faculty of Medicine, University of New South Wales, Sydney, New South Wales, Australia
¹²Sexual and Reproductive Health Unit, Papua New Guinea Institute of Medical Research, Goroka, Eastern Highlands Province, Papua New Guinea

**Acknowledgements** We thank the women who participated in this research and their families and community, without whom this work would not have been possible. Special thanks to the Papua New Guinea study team who collaborated to the fieldwork, and Cepheid (Sunnyvale, California, USA) for donating Xpert HPV cartridges.

**Contributors** MM: conceptualisation, methodology, laboratory work, writing original draft, software analysis. DAM: supervision, methodology, investigation, writing–review and editing. GT coordinated the evaluation of liquid-based cytology specimens, review and editing. SG: conceived the original trial, supervision, funding acquisition, project administration, writing—review and editing. SP: software analysis, data curation, review and editing. PB, DH and PJT: investigation, laboratory work, review and editing. SGB: conceived the original trial, coordinated clinical implementation, review and editing. JBo, JG, ZK, GM coordinated clinical implementation in the original trial, review and editing. AMC

and GH: investigation, review and editing. JBr, MS and JK conceived the original trial, funding acquisition, review and editing. AJV conceived the original trial, conceptualisation, funding acquisition, project administration, writing, review and editing. GLM: conceptualisation, supervision, funding acquisition, investigation, project administration, guarantor, writing, review and editing.

**Funding** This work was funded through the National Health and Medical Research Council, Australia, (NHMRC) grants 1013209 and 1104938 (AJV), Government of Papua New Guinea (ICRAS 297/1), the NHMRC Centre for Research Excellence in Cervical Cancer Control APP1135172 (JBr, MS, AJV, SG, JK, DAM and DH) and NHMRC Investigator Grant APP1197951 (SG).

**Competing interests** AJV, JG, JBo, GM, PJT, SGB and JK have received subsidised test kits for research from Cepheid. MS, JBr, GT and DH have received donated test kits for research from Abbott, BD, Cepheid, Hologic, Qiagen, Roche and Seegene. AJV and MS jointly lead the Elimination of Cervical Cancer in the Western Pacific (ECCWP) programme with philanthropic funding support from the Minderoo Foundation and the Frazer Family Foundation; and equipment, tests and consumables donated by Cepheid for HPV-based cervical screening in Papua New Guinea and Vanuatu. SG is a member of the Global Advisory Board for HPV vaccines Merck and has led investigator-initiated grants from Merck on HPV in young women. MM, DAM, SP, PB, GH, ZK and GLM declare no conflicting interests.

**Patient and public involvement** Patients and/or the public were not involved in the design, or conduct, or reporting, or dissemination plans of this research.

**Patient consent for publication** Not applicable.

**Ethics approval** This study involves human participants and approval of the trial was provided by the Medical Research Advisory Committee (MRAC) of the Papua New Guinea National Department of Health (approval number 17.36), the Institutional Review Board of the Papua New Guinea Institute of Medical Research (IRB 1712) and the Human Research Ethics Committee (HREC) of UNSW Australia (approval number HC17631). Written informed consent was obtained from all participants prior to enrolment. Participants gave informed consent to participate in the study before taking part.

**Provenance and peer review** Not commissioned; externally peer reviewed.

**Data availability statement** Data are available on reasonable request. Limited data may be available on reasonable request from the corresponding author, subject to ethical approval.

**ORCID iDs**
Monica Molano http://orcid.org/0000-0002-9167-1234
John Bolnga http://orcid.org/0000-0003-1214-9817
Andrew John Vallely http://orcid.org/0000-0003-1558-4822

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
