## [Reviewer comments · BMJ Open]

ARTICLE DETAILS

TITLE (PROVISIONAL)	Performance of CADM1, MAL and miR124-2 methylation as triage markers for early detection of cervical cancer in self- and clinician-collected samples: an exploratory observational study in Papua New Guinea
AUTHORS	Molano, Monica; Machalek, Dorothy; Tan, Grace; Garland, Suzanne; Balgovind, Prisha; Haqshenas, Gholamreza; Munnall, Gloria; Phillips, Samuel; Badman, Steven; Bolnga, John; Cornall, Alyssa; Gabuzzi, Josephine; Kombati, Zure; Brotherton, Julia; Saville, Marion; Hawkes, David; Kaldor, John; Toliman, Pamela; Valley, Andrew; Murray, Gerald L

VERSION 1 – REVIEW

REVIEWER	Aparajitha Vaidyanathan Northwestern University Feinberg School of Medicine, Hematology/Oncology
REVIEW RETURNED	26-Nov-2023

GENERAL COMMENTS	In this manuscript, Molano et al., describe an exploratory clinical study where they assess the use of host methylation markers as triage tools to predict high-grade squamous intraepithelial lesions in samples collected from hrHPV+ women in Papua New Guinea (PNG). The authors use methylation specific PCR to detect the methylation levels of MAL, CADM1 and miR124-2 in 44 hrHPV+ sample from women that participated in an HPV screen and same day treat trial in PNG. This study provides a promising start with preliminary data encouraging the set-up of larger clinical studies to determine the ultimate usefulness of these methylation markers as a detection tool for cervical cancer in low- and middle-income countries. However, there are some concerns listed below that need to be addressed to strengthen the conclusions of this manuscript. Major points 1. The Introduction section should include some background on histology, staging and grading of cervical cancer. The authors talk about the various cytological grades of the cancer throughout the manuscript, but do not introduce it properly in the introduction section. A lot of this information was properly written in the protocol v0.4 attached with the manuscript. It is important to include this in the manuscript as well.
--

	2. The authors are encouraged to include some detail on DNA methylation markers. The basic biology and their effectiveness as a diagnostic marker, and what is known about CADM1, MAL and miR124-2 in cancers. This can be discussed briefly in the introduction and expanded in the discussion section. 3. In the methods section- please clarify what gene was used as a sample housekeeping gene to normalize the methylation expression Ct values. 4. Under the statistical analysis (methods section)- please give an example as to how the calculations was done to determine sensitivity and specificity. It not entirely clear as described in the methods section. 5. Please edit the manuscript thoroughly.
--	---

REVIEWER	Christopher Hillyar University of Oxford
REVIEW RETURNED	19-Dec-2023

GENERAL COMMENTS	Dear Author, The manuscript, entitled “Clinical performance of CADM1, MAL and miR124-2 methylation as triage markers for early detection of cervical cancer in self- and clinician-collected samples in Papua New Guinea”, aimed to characterize the diagnostic utility of CADM1, MAL and miR124-2 methylation for detection of HSIL/SCC. Due to issues with selection of the reference standard, the results on the diagnostic accuracy of the biomarkers are viewed with a degree of caution, as they may be overestimates due to the use of cytology as the reference standard. The introduction, results and discussion need to be updated accordingly. However, the execution of the study methodology (apart from choice of reference standard and over-analysis of data in Figure 1 using multiple Wilcoxon tests) was adequate. Abstract:  1. Reads clearly. 2. Please report values for sensitivity and specificity for miR124/MAL plus HPV16/HPV18. 3. Is it really useful? Strengths and limitations:  4. The first point under study strengths is not a strength. If one over-specifies the scope of something eventually one will have inevitably created something that is “first-in-class”. 5. The second point is not necessarily a strength. An obvious limitation of using different cut-off points for the same sample type collected in different ways is that if the sampling method is not known (as may happen in real-world applications) the result of running the test using the incorrect cut-off is that it will be spurious or uninterpretable. If the cut-off is standardized no matter the collection method, then this problem is not an issue. Multiple cut-offs reduce the real-world utility of the test. 6. Point four is also not a strength of the study; it is a future direction. Instruction:  7. If HPV nucleic acid testing is more sensitive than cytology at detecting high grade-disease, then the selection of cytology as a reference standard has negative implications on the results for the sensitivity of the index tests. This highlights a major methodological weakness, which, therefore, will lead to significant overestimation of
---

	the sensitivity of the index tests, with the true underlying sensitivities and specificities of the index tests being much more uncertain (and potentially overestimated) compared to those reported in the literature using histology as a reference standard. Methods and materials: 8. Clear. Results: 9. Figure 1: The Kruskal-Wallis test is the appropriate test to assess if DNA methylation was significantly different between the three different phenotypes. Please note only MAL and miR124-2 are able to differentiate phenotypes on clinician-collected cervical samples based on methylation of these genes. Over analysis using multiple Wilcoxon tests, which results in greater probability of reporting significant results, should be discouraged and the results should be altered accordingly. Discussion 10. Index tests with high sensitivity and low specificity will result in over-investigation of women at low risk for developing cancer and result in potential harm to patients and place significant financial burden on healthcare systems, which will be untenable in low resource settings. 11. Because there are limitations associated with calculation of sensitivity and specificity using a suboptimal reference standard (cytology), it is inappropriate to claim that miR124-2/MAL in combination with HPV16/18 (or any other biomarker in this study) would result in fewer referrals than the current algorithm. 12. For the same reason, this study alone is inadequate for recommendation of miR124-2/MAL for large field studies. It would be more appropriate to evaluate the diagnostic accuracy of these biomarkers using histology as the index test, before committing more resources to large field studies. 13. Higher sensitivity of index tests compared to the current algorithm may increase, not reduce, referrals for colposcopy/treatment, especially if the increase in sensitivity is not offset by sufficiently greater improvements in specificity. Sensitivities of 90% with specificities of only 70% for miR124-2/MAL are likely to result in a larger number of inappropriate referrals, compared to the results in the manuscript for HPV16/18 alone, which yielded sensitivity of 52% and specificity of 81%. 14. Given the above significant limitations/challenges, I would temper the conclusion so that it states that miR124-2/MAL is a “potential” biomarker for prediction of HSIL/SCC. 15. Future work should not focus on large scale studies at this time, as more accurate characterization of the true diagnostic accuracy using a histological reference standard is absolutely required.
--	--

VERSION 1 – AUTHOR RESPONSE

Reviewer: 1

Comment	Response	Change in text
1. The Introduction section should include some background on histology, staging and grading of cervical cancer. The authors talk about the various cytological grades of the cancer throughout the manuscript, but do not introduce it properly in the introduction	Some additional information has been added to the introduction	The following addition: “Infection with high-risk HPV types (hrHPV) may cause cervical abnormalities graded

section. A lot of this information was properly written in the protocol v0.4 attached with the manuscript. It is important to include this in the manuscript as well.		histologically as cervical intraepithelial neoplasia (CIN, grades 1-3) or cancer (principally SCC or adenocarcinoma), or graded cytologically (using exfoliated cells from the cervix) as LSIL, HSIL or SCC.6”
2. The authors are encouraged to include some detail on DNA methylation markers. The basic biology and their effectiveness as a diagnostic marker, and what is known about CADM1, MAL and miR124-2 in cancers. This can be discussed briefly in the introduction and expanded in the discussion section.	We are constrained by word limits and feel this has been covered to a suitable extent in the introduction, as well as in the discussion (where specific performance parameters are cited from the literature). For example, the following is from the Introduction in the original manuscript submission: “Aberrant methylation is pivotal in cervical carcinogenesis; it causes changes in gene expression, faulty condensation, and chromosomal instability. ¹³, ¹⁴ Increased DNA methylation has been associated with hrHPV persistence ¹³ and correlates with increasing severity of cervical disease. ¹⁴, ¹⁵ DNA methylation markers such as CADM1, MAL and miR124-2 have shown good performance for the detection of CIN2+ among hrHPV+ women. ¹⁴, ¹⁶, ¹⁷ However, data on their performance has been based on studies conducted in high income settings and among highly screened populations. ^{18”}	No change

3. In the methods section- please clarify what gene was used as a sample housekeeping gene to normalize the methylation expression Ct values.	The information has been added in the methods section	The following addition: “As a reference, a qMSP for the housekeeping gene b-actin (ACTB) was performed. Samples with Ct values for ACTB of >35 were considered ACTB negative as this indicated poor sample quality due to insufficient DNA or inadequate bisulphite conversion. Target DNA methylation levels were normalized to the reference gene as described previously.”
4. Under the statistical analysis (methods section)- please give an example as to how the calculations was done to determine sensitivity and specificity. It not entirely clear as described in the methods section.	Further detail on this calculation has been added to the manuscript	The following text: “Sensitivity and specificity were determined for the following triage strategies...” Has been changed to: “Sensitivity [number of correct positives (i.e. positive for at least one marker)/ number of reference assay positives] and specificity (number of correct negatives/number of reference assay negatives) were determined for the following triage strategies...”
5. Please edit the manuscript thoroughly.	The authors have re-read the manuscript and improved the expression	Throughout

Reviewer: 2

Comment	Response	Change in text
Abstract		
2. Please report values for sensitivity and specificity for miR124/MAL plus HPV16/HPV18	This change has been made	The following sentence: “miR124-2/MAL plus HPV16/HPV18 or extended genotyping improved sensitivity for HSIL+, but decreased specificity.” Has been changed to: “miR124-2/MAL plus HPV16/HPV18 improved sensitivity for HSIL+ (95.2%, 95:CI 76.2-99.9), but decreased specificity (55.0%, 95%CI: 31.5-76.9)”
3. Is it really useful?	The text has been changed to remove the word “useful”	“miR124-2/MAL methylation is a potential triage strategy for the detection of HSIL/SCC in LMIC.”
Strengths and limitations		
4. The first point under study strengths is not a strength. If one over-specifies the scope of something eventually one will have inevitably created something that is “first-in-class”	This point has been modified to highlight the strength of studying self-collection in LMIC	“This study uniquely investigated host methylation markers in the detection of cervical HSIL in self-collected samples in a low-middle income country”
5. The second point is not necessarily a strength. An obvious limitation of using different cut-off points for the same sample type collected in different ways is that if the sampling method is not known (as may happen in real-world applications) the result of running the test using the incorrect cut-off is that it will be spurious or uninterpretable. If the cut-off is standardized no matter the collection method, then this problem is not an	We agree that having an incorrect cutoff is problematic. Our study and several others have shown that self-collected samples and clinician collected samples differ in epigenetic profile due to different cell composition. An important methodological feature of	“A direct comparison of self-collected samples and paired clinician-collected samples was possible, thus identifying different performance optimal cutoff values by sample type”

issue. Multiple cut-offs reduce the real-world utility of the test	this study is the ability to directly compare performance between self and clinician collected samples, which serves to highlight this difference. This point has now been modified to highlight the strength of having paired samples for a direct comparison by collection method.	
6. Point four is also not a strength of the study; it is a future direction	Dot point 4 has been removed	
Introduction		
7. If HPV nucleic acid testing is more sensitive than cytology at detecting high grade-disease, then the selection of cytology as a reference standard has negative implications on the results for the sensitivity of the index tests. This highlights a major methodological weakness, which, therefore, will lead to significant overestimation of the sensitivity of the index tests, with the true underlying sensitivities and specificities of the index tests being much more uncertain (and potentially overestimated) compared to those reported in the literature using histology as a reference standard.	We are aware of the potential weakness introduced by the use of cytology as a reference test. In the original submission this was canvassed in the “Strengths and Limitations” section, and a separate “study limitations” paragraph at the end of the discussion. Notably, risk of disease misclassification was minimised by slide assessment by two independent experienced cytologists, with dual p16/Ki-67 immunostaining used to resolve disagreements. Some of the text in the study limitations paragraph has been changed to indicate the potential for cytology	The following text from the study limitations: “It was not feasible to conduct cervical biopsies in this setting due to limited specialist staff and infrastructure; hence LBC HSIL was the diagnostic reference standard rather than a histological end point. This is unlikely to have led to disease misclassification as all slides were assessed by two independent experienced cytologists” Has been changed to: “It was not feasible to conduct cervical biopsies in this setting due to limited specialist staff and infrastructure; hence LBC

	errors to impact methylation marker performance. It is important to note that in the trial (and under normal clinical management) cytology does not play a role in patient care. Treatment is based almost exclusively on HPV test results. In this context applying a methylation test will help to reduce the unnecessary referral rate.	HSIL was the diagnostic reference standard rather than a histological end point and this may have impacted estimates of sensitivity/specificity for methylation markers. However, to avoid disease misclassification by cytology all slides were assessed by two independent experienced cytologists...”
Results:		
9. Figure 1: The Kruskal-Wallis test is the appropriate test to assess if DNA methylation was significantly different between the three different phenotypes. Please note only MAL and miR124-2 are able to differentiate phenotypes on clinician-collected cervical samples based on methylation of these genes. Over analysis using multiple Wilcoxon tests, which results in greater probability of reporting significant results, should be discouraged and the results should be altered accordingly.	The analysis by Wilcoxon test has been removed. The text has been modified to only refer to the Kruskal Wallis test results.	For example, the following: “In clinician-collected samples DNA methylation of MAL was significantly higher in HSIL (median 2.0%; 95%CI:0.7-20.4) and SCC (63.3%; 46.8-67.9) than normal/LSIL samples (1.4%; 0.8-3.1) (p=0.013 and p=0.004, respectively) (Figure 1, top central panel). Methylation of mirR124-2 was also significantly higher in HSIL (5.6%; 3.9-20.5) and SCC (24.1%; 12.4-34.5) than normal/LSIL samples (2.6%; 1.1-5.1) (p<0.001 and p=0.033, respectively) (Figure 1, top right panel). Methylation levels of both genes did not differ significantly between HSIL and SCC. Methylation of CADM1 did not differ significantly with increasing cytology grade (Figure 1, top left panel).”

		Has been changed to: “In clinician-collected samples DNA methylation of MAL was higher with disease grade from normal/LSIL samples (1.4%; 0.8-3.1), HSIL (median 2.0%; 95%CI:0.7-20.4) and SCC (63.3%; 46.8-67.9) (p=0.0046) (Figure 1, top central panel). Methylation of mirR124-2 was also higher with disease grade from normal/LSIL samples (2.6%; 1.1-5.1), HSIL (5.6%; 3.9-20.5) and SCC (24.1%; 12.4-34.5) (p=0.0015) (Figure 1, top right panel). Methylation of CADM1 did not differ significantly with increasing cytology grade (Figure 1, top left panel).”
Discussion		
10. Index tests with high sensitivity and low specificity will result in over-investigation of women at low risk for developing cancer and result in potential harm to patients and place significant financial burden on healthcare systems, which will be untenable in low resource settings.	We agree that this is a valid point. As mentioned above, the current primary test is for HPV (high sensitivity/low specificity), and treatment is based almost exclusively on this result. By applying a secondary (triage) methylation test, we can maintain the sensitivity and improve overall specificity of the detection of underlying disease.	
11. Because there are limitations associated with calculation of sensitivity and specificity using a suboptimal reference standard (cytology), it is inappropriate to claim that miR124-2/MAL in combination with HP16/18 (or any other biomarker in this study) would result in fewer referrals than the current	As mentioned above, it is important to note that in this study (and under normal clinical management) cytology does not play a role in patient care. Treatment is based almost exclusively	Changes made in discussion: “Addition of HPV16/18 or extensive genotyping will improve sensitivity at the

algorithm.	on HPV test results. In this context applying a methylation test can potentially reduce the unnecessary referral rate. References to the current algorithm have been removed from the manuscript	expense of lower specificity and higher referrals to immediate treatment, but still lower than using the current algorithm. “While the sensitivity/specificity was slightly lower than clinician-collected samples, this may be offset by higher screening rates achieved by this more acceptable method of collection and reduced referral to colposcopy/treatment compared to current screening algorithms.”
12. For the same reason, this study alone is inadequate for recommendation of miR124-2/MAL for large field studies. It would be more appropriate to evaluate the diagnostic accuracy of these biomarkers using histology as the index test, before committing more resources to large field studies.	A modification has been made along in the first paragraph of the discussion and in the conclusion. These now state evaluation against histology end points as a future step	e.g. The following text: “miR124-2/MAL methylation markers are suitable for evaluation in LMIC-based large field studies for triage of hrHPV+ women to identify high-grade disease.” Has been changed to: “miR124-2/MAL methylation markers are suitable for further evaluation against histology end points, and subsequently in LMIC-based field studies for triage of hrHPV+ women to identify high-grade disease.”
13. Higher sensitivity of index tests compared to the current algorithm may increase, not reduce, referrals for colposcopy/treatment, especially if the increase in sensitivity is not offset by sufficiently greater improvements in specificity. Sensitivities of 90% with specificities of only 70% for miR124-	In the study setting, treatment is provided immediately based upon HPV positivity, and in this context any additional triage will reduce referrals.	

2/MAL are likely to result in a larger number of inappropriate referrals, compared to the results in the manuscript for HPV16/18 alone, which yielded sensitivity of 52% and specificity of 81%.	In the scenario described, the testing of 16/18 alone would drop sensitivity by ~40% for a modest gain in specificity of ~10% compared to the miRNA124/MAL methylation test; in this case a lot of disease would be missed. Discussion of the implications of reduced specificity (e.g. in the context of self-collected samples) is already covered in the original submission.	
14. Given the above significant limitations/challenges, I would temper the conclusion so that it states that miR124-2/MAL is a “potential” biomarker for prediction of HSIL/SCC.		The conclusion has been changed from: “miR124-2/MAL methylation constitute a useful triage strategy to be investigated in a real-life ‘HPV screen-triage-and-same-day treat’ implementation program in LMIC.” To the following: “miR124-2/MAL methylation is a potential triage strategy for the detection of HSIL/SCC in LMIC.”
15. Future work should not focus on large scale studies at this time, as more accurate characterization of the true diagnostic accuracy using a histological	This has been indicated as the next step in further evaluation of these markers (in the Abstract,	

reference standard is absolutely required.	first paragraph of Discussion, and in the Conclusion)	
--	---	--

VERSION 2 – REVIEW

REVIEWER	Christopher Hillyar University of Oxford
REVIEW RETURNED	02-Apr-2024

GENERAL COMMENTS	Comments to author: The manuscript is generally well written, clearly structured, and the results and discussion are set nicely in context of the relevant literature. I recommend accepting the manuscript after minor revisions are addressed. Abstract: It might be more clear to add AUC before each AUC result. Otherwise very clearly written abstract. Strengths and limitations: Very clear and appropriate strengths and limitations cited. Introduction: The introduction is clear, concise and well structured. Please could you reference a paper by Verlaet which reported a three-gene methylation classifier (ASCL1, LHX8 and ST6GALNAC5) that was able to identify CIN3 in cervical self-samples with a ROC of 0.88. Verlaet W, Snoek BC, Heideman DaM et al. Identification and validation of a 3-gene methylation classifier for HPV-based cervical screening on self-samples. Clin. Cancer Res. 24(14), 3456–3464 (2018). Methods: The methods are clear, concise and well structured. Results: At line 308, please substitute “pattern” for “trend”, as this is statistically speaking more appropriate. At line 360, there is a grammatical error. Figure 2: It would be nice to see the AUC x-y plots and this allows a more impactful visual comparison of AUC data between biomarkers. Discussion: At line 426, there is a grammatical error. Please cite a reference to support the view that self-collection is the “more acceptable” method of specimen collection. Women may actually prefer clinicians to collect specimens, if they perceive that this mode of collection leads to more accurate diagnostic results from screening. Please discuss. Please discuss how the problem of HPV testing being more sensitive than the reference standard (cytology) at detecting high grade lesions, impacts on the utility of the results for all biomarkers included in your study that are based on HPV detection.
--

VERSION 2 – AUTHOR RESPONSE

Reviewer: 1

Comment	Response	Change in text
Abstract: It might be more clear to add AUC before each AUC result.	This has been amended in the abstract as suggested	Various parts in the Abstract, Page 3
Introduction: The introduction is clear, concise and well structured. Please could you reference a paper by Verlaat which reported a three-gene methylation classifier (ASCL1, LHX8 and ST6GALNAC5) that was able to identify CIN3 in cervical self-samples with a ROC of 0.88. Verlaat W, Snoek BC, Heideman DaM et al. Identification and validation of a 3-gene methylation classifier for HPV-based cervical screening on self-samples. Clin. Cancer Res. 24(14), 3456–3464 (2018).	The paper that the reviewer kindly suggested to add as a reference in the introduction was already referenced for us in the discussion but superficially. We really consider that this article is more valuable and interesting for the discussion. We have now extended the discussion improving the addition.	The following sentence in Discussion, line 465-466, Page 25 “Further work to identify markers suitable for self-collection should consider using self-collected samples for this purpose” Has been changed to: “Further discovery work to identify markers suitable for self-collection should consider using self-collected samples from LMIC. Verlaat et al., 2018 performed a chip-based genome DNA methylation screen on self-samples from a high-income country and identified a three-gene methylation classifier (ASCL1, LHX8 and ST6GALNAC5) that was able to identify CIN3 in cervical self-samples with a ROC of 0.88, which is superior to currently available methods”
Results: At line 308, please substitute “pattern” for “trend”, as this is statistically speaking more appropriate.	This change has been made	At line 311, Page 15 in Results, the sentence was changed: In self-collected samples, there was a trend of

		increasing DNA methylation of MAL...
At line 360, there is a grammatical error..	The correction has been made as suggested by the reviewer.	At line, 360-362, Page 18 in Results, the sentence was corrected: "HPV16/18/31/33/45/52/58 genotyping combined with different methylation markers showed high sensitivities at the expense of low specificities (Table 2)".
Figure 2: It would be nice to see the AUC x-y plots and this allows a more impactful visual comparison of AUC data between biomarkers	This change has been made as suggested in the Figure 2. We also included the 95% CIs in brackets next to the AUC.	Page 35. Legend of Figure 2 was modified: Receiver operating characteristic (ROC) curve values of the performance of DNA methylation of CADM1, MAL and miR124-2 for distinguishing HSIL from LSIL/normal, and SCC from LSIL/normal, stratified by type; clinician-collected cervical samples (A and C) or self-collected vaginal samples (B and D). In Results Page 14, line 307/309 was added: Figure 2 A, C In Results Page 15, line 319/321 was added: Figure 2 B, D
Discussion: At line 426, there is a grammatical error.		At line, 425-426, Page 23 in Discussion, the sentence was corrected: "CADM1 was not a good diagnostic marker, this is consistent with the findings of two other studies conducted in high-income countries"
Please cite a reference to support the	We added the reference in	At line, 433, Page 23 in

view that self-collection is the “more acceptable” method of specimen collection. Women may actually prefer clinicians to collect specimens, if they perceive that this mode of collection leads to more accurate diagnostic results from screening. Please discuss.	the text and discussed the point of the reviewer.	Discussion We add the reference number (11): Hawkes D, Keung MHT, Huang Y, et al. Self-Collection for Cervical Screening Programs: From Research to Reality. Cancers (Basel). 2020;12(4):1053
Please discuss how the problem of HPV testing being more sensitive than the reference standard (cytology) at detecting high grade lesions, impacts on the utility of the results for all biomarkers included in your study that are based on HPV detection.	The use of cytology as a reference standard will result in an overestimation of the sensitivity of the HPV test and decrease in the specificity, compared to using a histology reference. This will make it difficult to calculate the true underlying sensitivities and specificities of the HPV test and other molecular biomarkers. In the Papua New Guinea cohort, HPV testing has a high sensitivity for detection of HSIL but as reported by Vallely A et al, 2022 some overtreatment is observed. In our exploratory study by looking methylation as a triage marker on HPV positive women, the sensitivity is maintained and the specificity is improved, especially in clinician collected samples. However, these analysis have to be confirmed using histology data as end point of disease.	In the discussion, at line 480-482, Page 25, we added the following fragment: “Since HPV testing is more sensitive than cytology at detecting HSIL+, then the selection of cytology as a reference standard will result in an overestimation of the sensitivity of the HPV test and decrease in the specificity, compared to results using histology as a reference standard”.